# Novel Applications of Mesenchymal Stem Cell-Derived Exosomes for Myocardial Infarction Therapeutics

**DOI:** 10.3390/biom10050707

**Published:** 2020-05-02

**Authors:** Sho Joseph Ozaki Tan, Juliana Ferreria Floriano, Laura Nicastro, Costanza Emanueli, Francesco Catapano

**Affiliations:** 1National Heart and Lung Institute, Imperial College London, London W12 0NN, UK; sho.ozaki-tan17@imperial.ac.uk (S.J.O.T.); juliana.floriano@unesp.br (J.F.F.); l.nicastro19@imperial.ac.uk (L.N.); 2Botucatu Medical School, Sao Paulo State University, Botucatu 18618687, Brazil

**Keywords:** myocardial infarction, cardiovascular disease, mesenchymal stem cells, exosomes, extracellular vesicles, cardioprotection, cardiac regeneration, cell-free therapy

## Abstract

Cardiovascular diseases (CVDs) are the leading cause of mortality and morbidity globally, representing approximately a third of all deaths every year. The greater part of these cases is represented by myocardial infarction (MI), or heart attack as it is better known, which occurs when declining blood flow to the heart causes injury to cardiac tissue. Mesenchymal stem cells (MSCs) are multipotent stem cells that represent a promising vector for cell therapies that aim to treat MI due to their potent regenerative effects. However, it remains unclear the extent to which MSC-based therapies are able to induce regeneration in the heart and even less clear the degree to which clinical outcomes could be improved. Exosomes, which are small extracellular vesicles (EVs) known to have implications in intracellular communication, derived from MSCs (MSC-Exos), have recently emerged as a novel cell-free vector that is capable of conferring cardio-protection and regeneration in target cardiac cells. In this review, we assess the current state of research of MSC-Exos in the context of MI. In particular, we place emphasis on the mechanisms of action by which MSC-Exos accomplish their therapeutic effects, along with commentary on the current difficulties faced with exosome research and the ongoing clinical applications of stem-cell derived exosomes in different medical contexts.

## 1. Introduction

Myocardial infarction (MI) is a major cardiovascular disease (CVD) caused by a sudden full or partial stoppage of blood flow to the myocardium, leading to cardiomyocyte (CMC) death and subsequent irreversible heart muscle necrosis and apoptosis. While recent pharmacological and mechanical advances have significantly contributed to the sharp decline in death rates [1,2,3], MI continues to be a major cause of mortality and morbidity worldwide. In a global report on the incidence of disease and injury, it was estimated that around 10.6 million incidences of MI caused by ischaemic heart disease had occurred in 2019 alone [4].

The current standard treatments for MI that aim to mitigate heart damage remain limited in restoring heart function as they fail to address the CMC and vasculature loss underlying the condition [5]. Major strides in several biomedical fields—especially that of stem cell biology and medicine—have appreciably attracted attention towards the research and development of cardiac regeneration strategies following MI [6,7,8]. Thus, novel approaches for acute MI that stimulate angiogenesis, promote myocardial regeneration, and prevent left ventricular dysfunction have become highly sought after [8]. Over the past decade, a significant proportion of the clinical studies focused on cardiac regeneration have been centred around cell therapies, involving the engraftment of novel cellular agents including mesenchymal stem cells (MSCs) [9,10,11,12,13] and cardiac progenitor cells (CPCs) [14,15,16,17]. Whilst many of these studies have met clinical safety and risk standards, they have generally been unable to demonstrate significant benefits to cardiac function [18,19,20]. This failure has prompted a search for other novel approaches to induce cardiac regeneration, following acute MI.

Substantial evidence suggest that the observed cell therapy-related effects are attributed to the paracrine activity of the injected cells, rather than their successful integration/transdifferentiation into the myocardium [21,22,23,24,25,26,27]. Multiple reports have demonstrated that MSCs [28,29,30,31,32,33,34], embryonic stem cells (ESCs) [35,36,37,38], CPCs [39,40,41,42,43], and induced pluripotent stem cells (iPSCs) [44] mediate cardiac remodelling through such paracrine signalling activities. This paradigm shift has turned the direction of research towards the development of cell-free strategies focused on the isolation and application of these bioactive and pro-regenerative paracrine mediators. Studies have shown that extracellular vesicles (EVs)—particularly of the subset known as exosomes—are the premier paracrine factor responsible for promoting protection and regeneration in CMCs [44,45,46,47,48]. Specifically, MSC-derived exosomes (MSC-Exos) have been highlighted as potent paracrine vectors for reducing MI injury and rejuvenating cardiac function [47,48,49,50,51,52].

This review will examine the current state of MSC-Exo research and discuss its potential translational application as a novel cell-free agent for conferring cardiovascular protection and regeneration following MI.

## 2. Brief Overview of EVs and Exosome Biology 

EVs are prokaryote and higher eukaryote cell-secreted miniscule vesicles (30 nm to several micrometres in diameter) that act as shuttles for a heterogeneous and bioactive cargo mainly composed of proteins, lipids, and nucleic acids [53,54]. While initially regarded as membrane debris with no remarkable biological significance, EVs are today understood to be key agents of intercellular communication that are involved in the regulation of a diverse range of pathological and physiological processes.

Despite the umbrella term used to generally describe secreted phospholipid bilayer-enclosed vesicles, EV populations are known to be highly heterogeneous [54,55,56]. The characterisation and classification of EVs remain a translational hurdle, as complex overlapping physiochemical and biochemical properties between the different subtypes of EVs make their taxonomic organisation difficult to define in a rigorously discrete manner [57,58]. As a result, a reliable and universal EVs marker has not yet been identified. Furthermore, the vast range of cell-type-specific surface proteins represents an additional layer of complexity to exosome (and wider EV) classification. To date, EVs can be broadly divided based on their biogenesis into three subpopulations: apoptotic bodies, microvesicles, and exosomes (see Figure 1).

Apoptotic bodies (ABs) are the largest (800–5000 nm in diameter) subtype of EVs that are released as a product of apoptotic cell disassembly. Specifically, ABs release occurs during the execution phase of the apoptotic process, when the cell undergoes shrinkage, chromosomal condensation, nuclear and chromosomal fragmentation, and membrane blebbing [59]. While blebbing occurs, parts of the membrane and the cytoplasm separate from the cell to form ABs. It is thought that these ABs play an important role in the detection and removal of dead and dying cells via a variety of intercellular signalling pathways [59,60,61]. Microvesicles (MVs) are vesicular structures ranging from 100 to 1000 nm in diameter that are formed from outward blebbing of the plasma membrane of the parent cell. These MVs are known to be highly involved in intercellular communication through bioactive molecules shuttling [62].

Exosomes, which are the principal focus of this review, originate from the endocytic pathway. Like the larger MVs, exosomes play crucial roles in intercellular communications, but are divergently formed by the inward budding of the multivesicular body (MVB) membrane, which subsequently fuses with the plasma membrane to release intraluminal vesicles (ILVs) into the extracellular space [62]. As such, while MVs and exosomes are structurally similar, they greatly differ in size, lipid, and cargo composition [63]. Exosome-secreted vesicles are typically around 30–200 nm in diameter and selectively taken up by neighbouring or distant target cells. Upon receptor mediated uptake and/or internalisation, exosomes specifically modulate the recipient cell pathways according to the composition of their bioactive cargo [64]. Exosome membrane-associated proteins are enriched in tetraspanins (e.g., CD9, CD63, CD81, and CD82), MHC-I and MHC-II, heat-shock proteins (e.g., Hspa8, Hsp60, Hsp70, and Hsp90), GTPases (EEF1A1 and EEF2), and other proteins involved in MVB biogenesis (Alix and TSG-101) [65]. Additionally, their cargo contains many bioactive molecules, such as lipids [66], proteins [67], mRNAs, ribosomal RNAs (rRNAs), transfer RNAs (tRNAs), long noncoding RNAs (lncRNAs), microRNAs (miRNAs), and mitochondrial DNA (mtDNA) [68].

Although exosomal cargo specificity varies according to the parent cell-type and other environmental conditions (e.g., local temperature [69], O_2_ content [70,71], and pathological state [72,73,74]), there are numerous proteins highly associated with exosomes (including heat shock 70 kDa protein 8 (HSPA8), CD9, glyceraldehyde 3-phosphate dehydrogenase (GAPDH), beta actin (ACTB), CD63, CD81, annexin A2 (ANXA2), enolase 1 (ENO1), heat shock protein HSP 90-alpha (HSP90AA1), elongation factor 1-alpha 1 (EEF1A1), pyruvate kinase isozyme M2 (PKM2), 14-3-3 protein epsilon (YWHAE), syntenin-1 (SDCBP), programmed cell death-6 interacting protein (PDCD6IP), serum albumin (ALB), 14-3-3 protein zeta (YWHAZ), eukaryotic elongation factor 2 (EEF2), gamma actin (ACTG1), lactate dehydrogenase A (LDHA), heat shock protein HSP 90-beta (HSP90AB1), aldolase A (ALDOA), moesin (MSN), annexin A5 (ANXA5), phosphoglycerate kinase 1 (PGK1), and cofilin 1 (CFL1) [75]. The aforementioned characteristics, along with their unique mechanism of actions, make exosomes of immense biological interest, as testified by the plethora of studies aimed at employing them both as non-invasive diagnostic biomarkers [76,77,78,79,80] and as biological delivery systems [81,82,83,84].

### Exosomes: Techniques for Isolation and Characterization

Due to their ability to convey a biological message by shuttling a highly specific parental cell-derived cargo, including miRNAs, mRNAs, proteins, and a variety of bioactive molecules, EVs exert a pivotal role in cell-to-cell signalling. Hence, EV research is a growing interest worldwide and has led to a large number of clinical trials, aimed at investigating their efficacy and safety as cell-free therapy agents. Although EVs were discovered more than three decades ago, their isolation does not yet rely on a gold-standard protocol, but it is rather accomplished by different approaches, widely depending on their source (biofluids or cell culture media) and the following downstream applications [85,86].

High speed (100,000× *g*) ultracentrifugation (UC), the most commonly used method of isolation, allows low yield exosomes isolations from large volumes of initial material (usually urine or conditioned cell culture media) with the disadvantage of it being considerably time consuming and unable to separate EVs from other contaminants [87,88]. However, the recent commercialization of a variety of nanomembrane-based filters for ultrafiltration (UF) produces low contaminated samples in less than an hour. The resulting UC and UF pre-isolated EVs can be further purified through size exclusion chromatography (SEC) columns, which are made of porous resin particles. These columns remove most of the soluble proteins and other contaminants in the sample, providing highly purified exosome preparations [89]. Fast-isolation protocols utilizing a broad range of commercially available polymer-based precipitation buffers such as polyethylene glycol (PEG), have also been established with the aim of providing cost-effective and high yield exosome sample preparation in about 30 min. However, the high concentration of non-exosomal contaminants in the sample is a significant limitation of the precipitation-based columns that are therefore recommended for RNA/miRNA profiling only [90]. Recently, novel protocols relying on the asymmetric-flow field-flow fractionation (AF4) technology have been developed with the aim of separating label-free EVs at high resolution (1 nm) by analysing their hydrodynamic size [91,92].

Although exosome quantification remains challenging (due to their small size), transmission electron microscopy (TEM), Western blot (WB) for exosomal markers (CD9, CD63, CD81, and TSG-101), and nanoparticle tracking assays (NTA) can be used for post-isolation characterization of the exosomes [93]. Qualitative TEM protocols involving immunogold sample labelling can provide insights on exosome morphology as well as their surface protein composition. NTA analysis provides size, distribution and concentration by tracking the exosome’s Brownian motion [94]. Moreover, total protein quantification of the sample lysate can be obtained by micro-BCA assays. The above-mentioned methods are usually combined with WB analysis for exosome phenotyping. 

## 3. MSC-Exos in Cardiac Regeneration

MSCs are multipotent progenitor cells, which can be extracted from a wide variety of tissues, including bone marrow, adipose tissue, synovium, and Wharton’s jelly. While there exists some biological variability between MSCs isolated from different tissue origins, the International Society for Cellular Therapy has proposed a set of minimum criteria defining MSCs based on their characteristics: (a) being plastic-adherent when maintained in standard culture conditions; (b) expressing CD105, CD73, and CD90; (c) not expressing (negative markers) CD45, CD34, CD14, or CD11b, CD79a, or CD19 and HLA-DR surface molecules; (d) being able to differentiate into osteoblasts, adipocytes, and chondroblasts in vitro upon growth factor stimulation; and (e) being self-renewable, multipotent, easily accessible, and culturally expandable in vitro [95].

In addition, the genomic stability and lack of ethical issues with the application of MSCs make them exemplary vectors for cell therapy, regenerative medicine, and tissue repairment. Preclinical and clinical studies have reported generally positive immunomodulatory and regenerative effects of MSCs in various medical contexts, including cardiac regeneration. in vitro and in vivo studies have shown that MSCs are capable of differentiating into CMCs [96,97], endothelial cells (ECs) [98,99], and vascular smooth muscle cells (VSMCs) [100,101]. However, the beneficial effects of transplanted MSCs have proved to be modest and somewhat inconsistent. MSCs injections have been observed to consistently suffer from low engraftment and survival rates in recipient hearts [97,102,103], representing a daunting limiting factor for the development of cell-based translational solutions for MI. Studies observed rapidly declining cell count after MSCs transplantation, which indicated that observed reductions in infarct size following MSCs treatment are unlikely to be due to direct CMCs differentiation and repair [102,103]. It has been suggested that the observed decline in cell count may be attributed to the post-MI environment being inhospitable for cell survival [103].

Initial models attempting to provide mechanistic explanations of the therapeutic effects of MSCs described them as migratory cells travelling to and engrafting at the injury sites and subsequently interacting with local cells. However, recent studies have demonstrated that their therapeutic activity is mainly exerted in a paracrine manner rather than via direct stem-cell transdifferentiation. Such investigations have shown that MSC-conditioned medium enhances CMC and progenitor survival after hypoxia-induced injury [104,105]. This paracrine effect is facilitated by secreted exosomes. MSC-Exo cargo contains a variety of cytokines (e.g., IL-6 and IL-10 [106,107]), growth factors (e.g., TGF-β and HGF [99,100]), signalling lipids [108], mRNAs (e.g., *IGF-1R* [109]), and regulatory miRNAs (e.g., miR-21 and miR-133b [110,111]). These components play major and minor modulatory roles in a broad range of physiological processes, including organism development, epigenetic regulation, immunomodulation [112], tumorigenesis, and tumour progression [113]. Furthermore, the therapeutic applications of MSC-Exos provide multiple advantages over pure cell treatments, including negligible risk of tumour formation and significantly lower immunogenicity.

The above indicates strong potential implications of MSC-Exos in novel therapeutics for cardiovascular diseases. In fact, a large volume of preclinical studies has confirmed that MSC-Exos reduce the infarct size and improve post-MI cardiac function [50,51,52,114,115,116]. Specifically, blood flow recovery and preserved cardiac systolic and diastolic performance has been consistently observed [116].

### 3.1. MSC-Exos Increases Angiogenesis

Angiogenesis is the physiological process by which new blood vessels form and develop from existing vasculature. The post-MI myocardium suffers from a limited pro-angiogenic capacity [117]. Severe angiogenic impairment may contribute to contractile dysfunction following heart failure as the oxygen supply to the vasculature is depleted. Therefore, therapeutic solutions promoting *de novo* microvessels’ formation represent a promising strategy for the treatment of acute MI. MSCs contribute to cardioprotection and regeneration in an infarcted myocardium through its paracrine stimulation of angiogenesis in affected cells. Studies have shown that this pro-angiogenic potential is promoted by MSC-Exo-mediated secretion of bioactive factors (see Table 1) [118]. However, it remains unclear the extent to which MSC-induced angiogenesis can be attributed to MSC-Exo-mediated effects [119].

Significant blood vessel neo-formations including T cell proliferation and tube formation have been observed in vitro after MSC-Exos administration [120,121]. In parallel, expression analyses showed that a number of pro-angiogenic and angiogenesis-associated factors were significantly upregulated after MSC-Exos treatment in different cardiac cells. Particularly noteworthy was that numerous in vitro studies reported significant upregulation in ECs of vascular endothelial growth factor (VEGF), an essential component for maintaining vascular homeostasis and stimulating the angiogenic cascade [122,123,124,125,126]. Interestingly, in a recent study in which neonatal rat CMCs were treated with exosomes from different sources (bone marrow-derived MSCs (BM-MSCs), adipose tissue-derived MSCs (AD-MSCs), and umbilical cord-derived MSCs (UC-MSCs)), VEGF, pro-angiogenic fibroblast growth factor-β (FGF-β), and hepatocyte growth factor (HGF) levels were markedly increased in target cells [126]. Notably, AD-MSCs-exosomes had the most significant effect. In addition, hypoxic AD-MSC-Exo treatment promotes the upregulation of the pro-angiogenesis genes angiopoietin-1 (*Angpt1*) and receptor tyrosine kinase Flk-1 (*Flk1*) while simultaneously down-regulating the anti-angiogenesis genes vasohibin-1 (*Vash1*) and thrombospondin-1 (*Tsp1*) in ECs [127]. These expression changes were induced by the hypoxic AD-MSC-Exos-mediated activation of the protein kinase A (PKA) signalling pathway. A follow-up study demonstrated that PKA activation triggers VEGF expression in ECs, synergistically regulates *Ang1* and *Flk1* expression, and inhibits the expression of *Vash1* [128].

Although the mechanistic bases need to be further elucidated, it is clear that MSC-Exo-induced angiogenesis is strictly cargo-dependent. MSC-Exos exposed to ischemic conditions have a high representation of factors involved in canonical angiogenesis-associated pathways, such as the cadherin, epidermal growth factor receptor (EGFR), FGF, and platelet-derived growth factor (PDGF) pathways [119]. Further network analysis of the MSC-Exo-induced angiogenesis interactome showed that protein nodes (i.e., units in an analysis network that represent a specific protein) were most represented in clusters around canonical angiogenesis pathways such as nuclear factor kappa B1/2 (NF-κB1/2), avian reticuloendothelial virus oncogene homolog A (RELA), platelet-derived growth factor receptor-β (PDGFR-β), and EGFR in ECs [119]. In particular, MSC-Exo-induced angiogenesis in ECs is dependent on NF-κB signalling in a dose-dependent manner. Additionally, in ischaemic MSCs, the expression of a similar subset of angiogenic signalling pathways was also significantly increased. These findings suggest that ischaemic MSCs are able to create a pro-angiogenic environment via secretion of exosomes, thereby promoting *in situ* tissue healing [120].

Further proteomic studies reinforced the hypothesis that the aforementioned pro-angiogenic response is mediated by a consistent transfer of bioactive factors, such as the extracellular matrix metalloprotease inducer (EMMPRIN), matrix metalloprotease-9 (MMP-9), and VEGF between donor (MSCs) and recipient (ECs) cells [128]. Of particular interest is EMMPRIN, which mediates cell migration and angiogenesis upstream of MMPs and VEGF. Another study aimed at evaluating the molecular composition and the functional properties of the MSC-EV sub-populations found that numerous pro-angiogenic and pro-migratory molecules, including VEGF, transforming growth factor-β (TGF-β), interleukin-8 (IL-8), and PDGF factors and PDGFR-α/β, were compartmentalised in MSC-Exos [129]. A separate proteomic analysis showed that MSC-Exos contain galectin-1, ezrin, and p195, which are cell adhesion proteins associated with angiogenesis and cell proliferation [130].

In addition to their protein fraction, MSC-Exos are able to convey their pro-angiogenic signals through a direct miRNA transfer. Hypoxia-elicited MSC-Exos are significantly enriched in pro-angiogenic miR-125b-5p compared with naturally occurring MSC-Exos [131]. miR-21-5p is also enriched, leading to increased expression of the TGF-β signalling pathway, pro-angiogenic VEGF-α, ANGPT-1, hypertrophic atrial natriuretic factor (ANP), and brain natriuretic peptide (BNP) [132]. Another landmark study identified high levels of the pro-angiogenic miR-21, miR-1246, miR-23a-3p, and miR-23, in MSC-Exos [133]. It was subsequently discovered that a set of angiogenesis-associated genes, including members of the angiopoietin network (ANGPT1, ANGPT4, and ANGPTL4) as well as other important mediators of angiogenesis (ephrin type-B receptor 2 (EPHB2), and neuropilin 2 (NRP2)), were upregulated in MSC-Exos treated ECs. Furthermore, numerous genes that correlate with VEGF or increase its expression, such as MYC-associated zinc finger protein (MAZ), semaphorin 5B (SEMA5B), and nuclear receptor coactivator 1 (NCOA1), were also significantly upregulated. In contrast, several antiangiogenic genes including Serine protease inhibitor Kazal-type 5 (SPINK5), arachidonate 5-Lipoxygenase (ALOX5), and protein phosphatase 1A (PPM1A) were significantly down-regulated in MSC-Exo-treated ECs [133].

### 3.2. MSC-Exos Reduces Apoptosis 

Apoptosis is a form of programmed cell death characterised by membrane blebbing, cell shrinkage, condensation of chromatin, and DNA fragmentation, followed by a rapid engulfment of the corpse by neighbouring cells. It is distinguished from necrosis by the absence of an associated inflammatory response [156]. Apoptosis plays a significant role during acute MI and reperfusion-induced tissue injury, leading to the myocardial loss that eventually manifests as heart failure [157,158]. Therefore, suppressing apoptosis in CMCs is potentially an effective strategy in the alleviation of acute myocardial infarction [159]. MSC-Exos confer cardioprotection under ischemic conditions to cardiac cells by hypoxia-induced apoptosis inhibition (see Table 2). This anti-apoptotic response hinders myocardial damage, preserves left ventricle geometry, and improves cardiac function [160].

Evidence suggests that MSC-Exos exert anti-apoptotic effects through the modulation of bioenergetics in target cells. Key features of acute MI include loss of ATP and NADH, increased oxidative stress, and cell death [161]—all processes that are directly tied to cellular bioenergetics. MSC-Exo treatment in murine myocardium increases ATP and NADH levels, decreases oxidative stress, and enhances phosphoinositide 3-kinase (PI3K)/protein kinase B (AKT) pro-survival signalling activation in ischemia-reperfusion injury (I/R) hearts [51]. In parallel, MSC-Exo treatments reduce c-Jun N-terminal kinase (c-JNK) phosphorylation, a major activator of pro-apoptotic signals. This finding suggests that MSC-Exos exert therapeutic effects at least partly through the restoration of bioenergetics in target cardiac cells. While the underlying anti-apoptotic molecular mechanisms remained unclear, it was speculated that MSC-Exos deliver a set of oxidative enzymes that are lost during I/R injury [161]. Consequentially, a restoration of the bioenergetics processes and simultaneous decrease in oxidative stress results in apoptosis reduction.

The PI3K/AKT pathway is a key intracellular signal transduction pathway involved in the regulation of apoptosis and survival. AKT is the primary protein effector downstream of the PI3K signalling pathway and plays an important role in glucose metabolism by regulating the biological functions of insulin [162,163]. CMC apoptosis is increased by malfunction of the AKT signalling pathway during hyperglycaemia, which is accompanied by an increase in the release of cytochrome *c* from mitochondria and an enhancement of caspase-3 activity [164]. This pathway is tightly regulated by phosphatase and tensin homolog (PTEN) via its phosphatase dephosphorylation activity [165]. Among the MSC-Exo anti-apoptotic miRNAs modulating the PI3K/AKT pathway, miR-144, which is highly enriched in BM-MSC-Exos, significantly counteracts apoptosis in hypoxic CMCs by interacting with PTEN/PI3K/AKT [166]. Similarly, miR-486-5p from BM-MSC-Exos reduce MI-induced apoptosis by repressing the PTEN pathway and subsequently activating the PI3K/AKT pathway in CMCs [167]. These observations are consistent with other studies investigating the effects of non-exosomal interventions on the PI3K/AKT pathway and apoptosis in the heart [168,169,170].

MSC-Exos treatment activates AMPK/mTOR and AKT/mTOR signalling to partly reduce in vitro and in vivo apoptosis through autophagy enhancement [171]. It is likely that PI3K/AKT activation contributes to the aforementioned anti-apoptotic response by accelerating autophagic signalling pathways. In addition, AD-MSC-Exos counteract apoptosis by Wnt/β-catenin pathway modulation, a key regulator of survival in CMCs [172]. AD-MSC-Exo treatments induce Wnt/β-catenin signalling activation by attenuating I/R- and hypoxia-reoxygenation injury (H/R; an in vitro model where standard culture atmosphere is replaced with a hypoxic or anoxic gas mixture to study the recovery process following the hypoxic period)-induced inhibition of Wnt3a, p-GSK-3β (Ser9), and β-catenin expression. This effect coincided with dramatically reduced I/R-induced apoptosis in rat CMCs, upregulation of Bcl-2 and cyclin-D1, downregulation of Bax and, inhibition of caspase-3 activity [172].

Hypoxic preconditioning of parent MSC significantly improves the ability of MSC-Exos to inhibit apoptosis by enriching their miR-22 content, which inhibits apoptosis by targeting methyl CpG binding protein 2 (Mecp2) [173]. Horizontal transfer of miR-22 reduces apoptosis in CMCs, ameliorates fibrosis, and improves post-MI function in the mouse heart. Likewise, miR-21 transfer via MSC-Exos enhances cardioprotection by conferring anti-apoptotic effects [123]. miR-21 is involved in several intracellular signalling pathways and modulates apoptotic proteins in CMCs, such as PDCD4, TLR4, NF-κB, and, notably, PTEN/AKT/Bcl-2 [174]. One study even found that miR-21 from BM-MSC-Exos protects cardiac stem cells expressing the stem cell factor receptor c-kit from oxidative injury and apoptosis through PTEN/PI3K/AKT pathway modulation [175]. Hypoxic MSC-Exos inhibit CMC apoptosis after acute MI by upregulating miR-24 in target cells, which in turn inhibits apoptosis in murine CMCs [176] by repressing Bcl-2-like protein-11 (BIM) translation (a member of the B cell lymphoma-2 (Bcl-2) family of apoptosis-mediating proteins). Hypoxic MSC-Exos also facilitates cardiac repair via miR-125b-5p cargo activity following MI [131].

Interestingly, induced changes to gene expression in parent MSCs influence the anti-apoptotic properties of MSC-Exos. GATA-binding protein-4 (GATA-4) overexpression leads to increased growth factor release and EC-mediated angiogenesis [177]. Another study also found that GATA-4 regulates the expression of the members of the miR-15 family in MSCs and improves their survival in ischemic environments [178]. MSCs overexpressing GATA-4-derived exosomes (MSC^GATA-4^-DEs) expressing anti-apoptotic miRNAs, reduce apoptosis and preserve mitochondrial membrane potential in targeted hypoxic CMCs [179]. Additionally, miR-19a and miR-451 are highly expressed in CMCs treated with MSC^GATA-4^ -DEs. Further analysis showed that miR-19a downregulates PTEN and BIM expression resulting in AKT and ERK signalling pathways activation while inhibiting JNK/caspase-3 activation by targeting the transcription factor SRY-box transcription factor-6 (SOX-6) [180]. These observations suggest a central role of miR-19a in mediating the anti-apoptotic effects of MSC-Exo treatments. A prior investigation of MSC^GATA-4^ found that the cardioprotection induced by MSC^GATA-4^-DEs is partially mediated by miR-221 [181], which inhibits p53 modulator of apoptosis (PUMA), a pro-apoptotic member of the Bcl-2 protein family.

Furthermore, exosomes from MSCs treated with macrophage migration inhibitory factor (MIF) (MSC^MIF^) enhances myocardial repair by ameliorating the heart function, reducing heart remodelling, mitochondrial fragmentation, and apoptosis in vivo [182]. These MSC^MIF^-DEs confer enhanced anti-apoptotic effects compared to unmodified MSC-Exos in hypoxic CMCs. Here, the exosomal long coding RNA lncRNA-NEAT1 inhibits miR-142-3p [183]. Increased activity of miR-142 has been found to induce apoptosis and cardiac dysfunction while its reduction rescued cardiac function in a murine heart failure model [184]. Additionally, activation of the lncRNA-NEAT1/miR-142 axis enhances the transcription factor forkhead box protein O1 (FOXO1) activity in CMCs, resulting in the downstream modulation of a wide range of genes regulating cellular apoptosis. It was therefore suggested that the lncRNA-NEAT1/miR-142/FOXO1 represents a novel cardioprotective signalling pathway.

### 3.3. Immune Response in Acute MI

Acute MI triggers inflammatory responses, which are in turn responsible for the healing/repair cycles following MI [190]. The role of reperfusion-induced inflammation in the repair process has been reported in several experimental models [191,192]. Initially, infiltrating monocytes and mast cells mediate cardiac repair by a complex process involving different cytokines and growth factor cascades [193]. However, a prolonged inflammation extends myocardial injury, leading to adverse left ventricular remodelling and heart failure [190]. In addition to the *in situ* inflammation affecting the infarcted region, acute MI also triggers systemic inflammation by modulating the levels of a wide range of immune factors involved in the humoral and cell-mediated inflammatory responses [194].

In particular, the complement cascade signalling is activated, mediating immune cell recruitment to the injured myocardium to rapidly elevate myocardial inflammation [190]. Mechanistically, this process infiltrates neutrophils and monocytes into the afflicted regions, where they exert different immune functions involved in inflammation and tissue repair such as degranulation, phagocytosis, and differentiation. In parallel, recruited mast cells degranulation activates a series of pro-inflammatory cytokine and chemokine cascades such as tumour necrosis factor (TNF), IL-1β, and members of the IL-6 family [194]. A variety of damage-associated molecular patterns (DAMPs) are simultaneously released from necrotic cardiac resident cells following infarction, which perform a variety of pro-inflammatory functions, including the activation of immune cells, TLR activation, and inflammasome formation signalling [195,196,197]. Finally, reactive oxygen species (ROS) generated from acute MI directly injures cardiac myocytes and vascular cells by triggering inflammatory cascades in a positive feedback loop [198,199].

Chronic and excessive pro-inflammatory response following acute MI contributes towards the induction of a process known as adverse left ventricular remodelling [194,200], which involves enhanced protease activation [201], cytokine expression [202], ventricular dilation [203,204], and excessive fibrosis induced by cardiac fibroblast activation [205]. This process is strongly correlated with worsened clinical outcomes, therefore making therapeutic targeting of inflammation following MI, an important strategy for limiting the infarct size.

#### 3.3.1. MSC-Exos Modulate the Immune Response

While the complete mechanism of action is not yet fully understood, MSCs have long been known for their immunomodulatory properties [206,207,208]. As the significance of their paracrine signalling is increasingly emphasised, evidence that MSCs alter the immune response via exosome shuttling during MI has recently emerged (see Table 3). Specifically, MSC-Exos hold potent immunosuppressive anti-inflammatory effects [120,209,210,211], having significant implications on cardiac tissue regeneration. Previous studies suggest that the switch from a pro-inflammatory to a tolerogenic immune response may contribute towards a pro-regenerative environment, allowing endogenous stem and progenitor cells to successfully repair the affected tissues [211].

A breakthrough study published in 2015 showed that MSC-Exos restrain the inflammatory response during acute MI by inhibiting immune cell invasion and proliferation in the infarcted zone of the rat heart [120]. Specifically, CD3^+^ T cells were significantly decreased after MSC-Exos treatment and a simultaneous inflammation/infiltration reduction was observed in myocardial tissue. Prior in vitro studies investigating the interaction between MSC-Exos and peripheral blood mononuclear cells (PBMCs) have also indicated similar results, as co-culture experiments increase CD3^+^ T cell apoptosis while reducing B cell proliferation, differentiation, and production of IgM, IgG, and IgA under CpG stimulation [212]. Furthermore, concentrations of immunosuppressive IL-10 were greatly increased in surrounding culture medium. It was subsequently reported that MSC-Exos inhibited the differentiation, activation, and proliferation of T cells in vitro in a similar manner [213].

Later studies aimed at identifying the mechanism by which MSC-Exos reduced the inflammatory response found numerous components involved in the process. MSC-Exos express programmed death-ligand 1 (PD-L1), galectin-1, and membrane-bound TGF-β, which are all essential molecules for inducing immune tolerance [214]. Moreover, immunomodulatory miR-29 and miR-24 are highly expressed in both MSC-Exos and MSCs [110], suggesting that MSC-Exos may be able to at least partially reproduce the immunosuppressive effects observed after MSC treatment. Both miRNAs modulate the immune response, with miR-29 being associated with reduced fibrosis via the repression of numerous collagen genes [215], and miR-24 limiting aortic vascular inflammation by interacting with a potent regulator of inflammation and tissue remodelling known as chitinase-3-like protein-1 (CHI3L-1) [216]. Additionally, miR-181a delivered by MSC-Exos inhibits the inflammatory response through c-Fos protein interaction, a key immunoactivator promoting the dendritic cell-related immune functions [217].

MSC-Exo-delivered miR-182 attenuates myocardial I/R injury via TLR4/NF-κB/PI3K/AKT pathway interaction, which in turn regulates macrophage polarisation [218]. MSC-Exo treatments modify the polarisation of pro-inflammatory M1 macrophages to anti-inflammatory M2 macrophages both in vitro and in vivo [218,219,220,221,222,223,224]. As opposed to the host-defensive regulatory M1 macrophages, M2 macrophages have functions in translating the pro-inflammatory cascades into reduced inflammatory cascades while enhancing subsequent reparative activities. MSC-Exo synthesis is affected by pro-inflammatory environments, which induce them to promote M2 polarisation by significant upregulation of miR-34a-5p, miR-21, and miR-146a-5p [223]. Besides the context of acute MI, the switch from an M1 to M2 phenotype has been similarly observed in MSC-Exo treatments in other physiological contexts, including bronchopulmonary dysplasia [221], cecal ligation, puncture-induced sepsis [222], and skeletal muscle damage [224].

Moreover, the M1 to M2 macrophages shift has implications in tissue fibrosis. The post-MI myocardium is highly characterised by extensive cardiac fibrosis due to the fibroblast and myofibroblast-mediated reparative response to ischemic cell death [225]. The initial cardiac fibrotic response is necessary to lessen the rupture of the ventricular wall and therefore represents an important step in prevention of subsequent heart failure [225]. The transition from acute inflammation to fibrosis, facilitated by the switch from M1 to M2 dominant macrophage subsets, significantly contributes towards increasing cardiac fibrosis.

M2 macrophage secretion of TGF-β1 induces resident fibroblast to myofibroblast transformation [226], which in turn produce matrix components more effectively. Furthermore, macrophages have roles in recruiting these cells to the injury site via chemokine signalling, and the resulting production of matrix components and collagen deposition stabilises and crosslinks to form scar tissue [227]. M2 macrophages further promote fibrogenesis through the production of arginase, which activates glutamate and proline, both of which are necessary for collagen synthesis [227,228]. It should be noted that acute MI often induces exaggerated response outside the injured area, which can contribute to progressive impairment of cardiac function and lead to heart failure. Conversely, M2 macrophages are a potent source of anti-inflammatory IL-10, which exerts protection against cardiac fibrosis. It has been shown that a lack of IL-10 leads to adverse tissue remodelling and more severe cardiac fibrosis when compared to naturally occurring counterparts [229].

Despite the seemingly contradictory functions of the M2 macrophages, several reports show that MSC-Exos ameliorate fibrosis after MI [114,173,210,230]. While the underlying mechanism is unclear, some studies have implicated a role for the miRNA component of the MSC-Exo molecular cargo. Ischemic preconditioned MSC-Exos contains miR-22, which ameliorates fibrosis and improves cardiac function post-MI [173]. MSC-Exos from miR-133-overexpressing MSCs produced comparable results [231,232].

## 4. Clinical Trials Involving Stem-Cell Derived Exosomes

Due to their potential as therapeutic cell-free drugs and biomarkers, 151 clinical trial studies involving exosomes [246] are being developed to-date. Although the regenerative and immunomodulatory activities of MSC-Exos have been shown in a *plethora* of preclinical studies in cardiovascular disorders, more efforts are needed to establish standard and consistent methods for exosome production (cell lines and culture conditions), isolation, and storage, which are aimed at reducing the variability between cell-free products. Additional investigations focused on dose and administration of exosome preparations in patients may facilitate their use in future clinical trials. Hence, due to the above translational limitations, in a majority of the ongoing clinical trials, exosomes and their protected cargos including RNAs, small RNAs, and proteins [247], are being investigated for developing novel diagnostic and prognostic tools toward broad range of conditions. Only seven interventional studies are currently evaluating the therapeutic efficacy and safety of stem-cell derived exosomes in patients (see Table 4).

### 4.1. MSC-EVs and Bronchopulmonary Dysplasia

The NCT03857841 phase I trial assesses the safety and tolerability of intravenous administration of BM-MSC-EVs (UNEX-42) in 18 patients at risk of bronchopulmonary dysplasia, a severe neonatal lung injury [248]. Preclinical studies in the hyperoxia (HYRX)-induced BPD model showed that MSC-EVs treatment improved lungs morphology by reducing fibrosis and inflammation. As extensively described in Section 3.3.1, MSC-EVs modulates macrophages phenotype by promoting the M1-like to M2-like status [221].

### 4.2. MSC-EVs in Dystrophic Epidermolysis Bullosa 

Epidermolysis Bullosa (EB) includes a group of inherited genodermatoses caused by a lack of collagen VII, which in turn results in skin fragility and mucocutaneous blistering. The phase I/IIA trial NCT04173650, aims at studying effectiveness and safety of AGLE-103 topic administration in the treatment of lesions in EB subjects. AGLE-103 is an allogenic derived drug composed of MSC-EVs derived from normal donors. Previous data showed that the intradermal administration of allogenic MSCs improved the wound healing and prevented blistering by promoting Collagen VII replacement in patients [249].

### 4.3. MSC-EVs in Patients with Acute Ischemic Stroke

The NCT03384433 phase II trial aims at studying efficacy and safety of allogenic miR-124 enriched MSC-EVs intravenously administered to five patients with acute ischemic stroke. Specifically, five patients aged 40–80 years will receive 200 µg (total protein) of miR-124-enriched allogenic MSC exosomes one month after stroke attack. 

MiR-124 regulates several biological processes in central nervous system [250] and exerts an anti-apoptotic and neuroprotective action in stroke [251]. In addition, previous studies showed that miR-124 promotes neurovascular remodelling, and neurogenesis after stroke [252].

### 4.4. Effect of MSC in Patients with Chronic Graft-Versus Host Diseases 

Chronic graft-versus-host disease (GVHD) is a life-threatening complication following allogenic hematopoietic stem cell transplantation, resulting in enhanced inflammatory events triggered by the interaction between donor lymphocytes and foreign antigens [253]. Major ocular complications including keratoconjunctivis, pain, photophobia, dryness and blindness, and other manifestations affecting the lacrimal glands, results in decreased tear production [254].

Participants from NCT04213248 Phase II clinical trial will receive umbilical mesenchymal stem cell (UMSC)-derived exosomes 10 µg/drop, four times a day for 14 days with the aim of relieving the dry eye symptoms. Previous results indicate that MSCs infusions can be therapeutically effective in suppressing dry eye symptoms associated in cGVHD subjects by promoting the generation of regulatory T cells exerting immunomodulatory effects [255]. Subsequently, Lay et al., confirmed that the anti-inflammatory MSCs capability is due to a paracrine action mediated by MSC-derived exosomes promoting IL-10-expressing regulatory cells proliferation and IL-17-expressing pathogenic T cells inhibition [256].

### 4.5. MSC-EVs Promotion of Macular Holes Healing

The early phase I trial NCT03437759 aims at inducing functional recovery of large and refractory macular holes (MHs). MHs are thickness defects in the eye macula region of various pathogenetic origin (including idiopathic and traumatic) [257].

Forty-four subjects will receive a 10 μL PBS drop containing 50 μg or 20 μg MSC-Exo into vitreous cavity and will be followed up by physical examinations, fundoscopy, best-corrected visual acuity (BCVA) measurement, and spectral-domain optical coherence tomography (OCT). The main patient inclusion criteria are a diagnosis of MHs larger than 400 μm. Literature studies show that local human umbilical cord-derived MSC-EVs injection ameliorates uveitis in a rat by inhibiting inflammatory cell migration, including neutrophils, macrophages cells, and CD4^+^ T cells [258].

### 4.6. MSC-EVs Treatment in Patients with Metastatic Pancreas Cancer 

Pancreatic ductal adenocarcinoma is a highly metastatic disease associated to KRAS Proto-Oncogene, GTPase (*KRAS*) gene mutations, occurring to > 90% of the patients [259]. In particular, the specific KRAS G12D mutation, a codon-12, exon-2 G > A point mutation subtype is the most negative prognostic factor associated with the modulations of both: the cell cycle regulator PI3K/AKT and, the cell proliferation/survival/regulation MEK pathways [260].

Twenty-eight subjects enrolled in the NCT03608631 study will receive different doses of iExosomes. Specifically, iExosomes is a preparation of MSC-EVs containing combined with KrasG12D siRNAs and the aim of this phase I clinical trial is to identify both dose-limiting toxicities (DLT) and maximum tolerated dose (MTD). The ability of iExosomes to enter cells via micropinocytosis and reducing RAS pathway activation, resulting in pancreatic cancer suppression, has been extensively shown in multiple in vitro and in vivo models [261]. RAS pathway is involved in a variety of cellular processes that are relevant for tumorigenesis [262].

### 4.7. MSC-EVs Based Treatment for Type I Diabetes Mellitus 

Type I diabetes mellitus (T1DM) is caused by the pancreatic islets β-cell damage, due to an autoimmune mechanism involving T cells [263] and other factors, including TNF-α and interferon-γ [264]. The rationale of the study is that umbilical cord-blood MSC-EVs intravenously infused can reduce the inflammation and stabilize the glycaemic control in T1DM patients by improving the β-cell mass. It has been previously shown that umbilical cord-blood MSC-EVs promotes the islet survival by reducing β-cell apoptosis [265]. In the phase III NCT02138331 clinical trial, twenty patients will receive a dose of 1.22–1.51 × 10^6^/kg of purified exosomes followed by a second injection of an equivalent dose of MVs (180–1000 nm) after 7 days.

## 5. Discussion and Future Perspectives

While there is still much to be understood about their nature, MSC-Exos have emerged as a highly promising cell-free vector for conferring regeneration in the heart. As outlined in this review, MSC-Exos may have especially strong implications in MI therapeutics, being able to confer potent angiogenesis [118,119,120,121], protection against apoptosis [160], and immunomodulation [120,209,210] that can directly counteract the adverse outcomes of MI and induce subsequent cardiac regeneration (see Figure 2). It is clear that the molecular cargo of MSC-Exos (especially that of miRNA and proteins) is highly responsible for the dynamic cardioprotective and regenerative effects observed.

While naturally occurring MSC-Exos have been demonstrated to induce desirable effects, it is a natural next step to pursue research of engineered exosomes that are capable of producing more vigorous effects that optimise cardioprotection and cardiac regeneration. Future research on MSC-Exos in this context should therefore focus on identifying the specific molecular cargo of MSC-Exos, and subsequently elucidate the mechanism by which such components are able to produce specific effects in target cells. Alongside this, research into the mechanism of exosome-loading process will be highly valuable as greater understanding of the process would allow for modification of the parent cell, with the aim of producing exosomes encapsulating desirable molecular cargo. While still unrefined, several aforementioned studies in this review have attempted to artificially improve MSC-Exo cardioprotection and regeneration, either through modification of the cells’ local environment [124,131,160,173,176] or by modifying the expression of specific genes that were believed to have downstream implications in cardioprotection and/or regeneration [119,177,178,181,183,184]. Furthermore, the prospect of developing synthetic exosome-mimics introduces the possibility of large-scale synthesis of MSC-Exo-like exosomes [266]. In this review, we provided an extensive list of specific molecular cargo components found in MSC-Exos that we hope can assist in designing such engineered exosomes.

Besides exosome modification, one interesting possibility is to explore the idea of the enhancement of cardioprotection and cardiac regeneration through selection of MSCs that secrete the “perfect” exosomes for the required clinical applications. To do this, cells could be subjected to undergo a process of directed evolution. This is a method for artificially selecting a biological species (usually a protein or nucleic acid) towards a user-defined goal [267]. It consists of subjecting a gene to iterative rounds of mutagenesis, selection, and amplification to create a product that is optimised to accomplish a specific goal—but this has never been attempted at the cellular level. In this logic, instead of rationally designing exosomes optimised to heal cardiac tissue, in vitro experiments attempting to select for MSCs that secrete exosomes best suited for inducing cardioprotection could pose as an alternative solution. Once such a population of exosomes is identified, their contents and mechanisms of action could then be analysed.

Nevertheless, the application of MSC-Exos—and exosomes in general—in a clinical setting remains a challenge due to limitations in exosome isolation and characterisation. While techniques such as ultracentrifugation and size exclusion chromatography-based methods are readily able to capture exosomes from MSC-conditioned media, it remains a hard truth that the resulting sample is never a pure population of exosomes due to the fact that other biological particles, especially that of MVs, can overlap in size. Developments of methods that can better distinguish MSC-Exos based on their biological properties rather than their physical qualities is likely to improve the pharmacological potency of the samples. More importantly, however, is that obtaining pure fractions of MSC-Exos would shed light on the components of the exosomes preparation that may produce unexpected and/or undesirable effects when applied in a clinical setting (such as an adverse immune response or tumour development) [268].

## Figures and Tables

**Figure 1 biomolecules-10-00707-f001:**
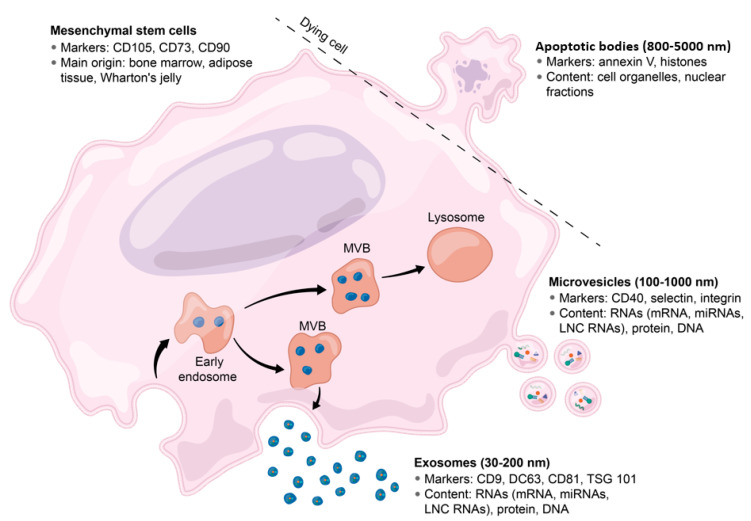
Biogenesis and features of apoptosis bodies (ABs), microvesicles (MVs), and exosomes. Each of the three classes of extracellular vesicle (EV) are characterised by unique biogenesis pathways, molecular markers, and cargo. ABs emerge as the cytoskeleton of apoptotic cells are disassembled and contain various cellular components such as organelles and nuclear fractions. MVs contain a variety of molecular cargo, including DNA, RNA, and protein, and are produced by the cell membrane budding and fission. Exosomes are the smallest class of EV, and are enriched in DNA, RNA, and proteins, and originate from the multivesicular body (MVB).

**Figure 2 biomolecules-10-00707-f002:**
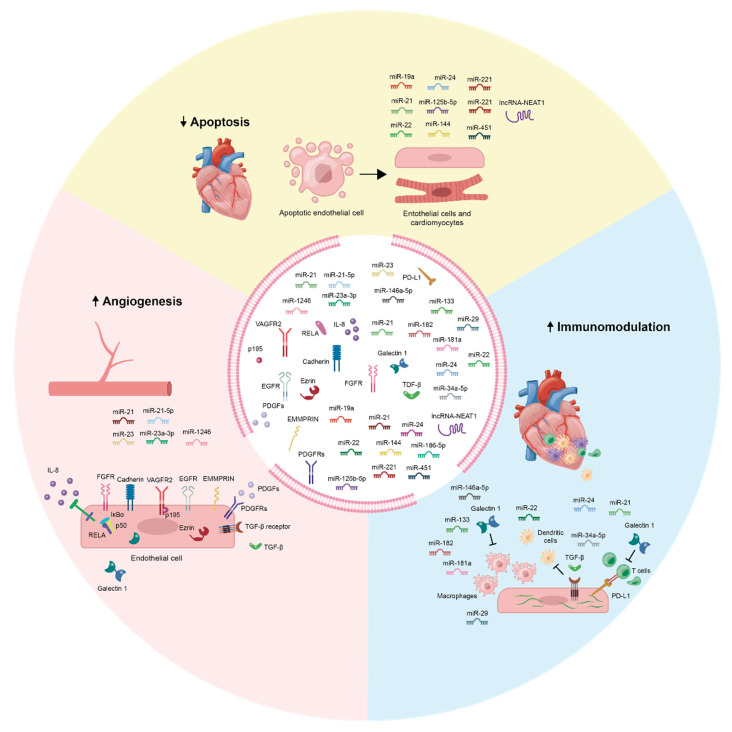
Summary of MSC-Exo molecular cargo and its functions in angiogenesis, apoptosis, and immunomodulation. MSC-Exo treatments improve blood vessel neo-formations through the activation of a wide range of pro-angiogenic pathways in ECs. In parallel, anti-apoptotic effects are induced via bioenergetics modification, principally though the PI3K/AKT and mTOR pathways. Finally, MSC-Exos modify the inflammatory and fibrotic immune responses, creating a microenvironment more accommodating to regeneration and healing. A highly diverse set of molecular cargo (almost solely consisted of miRNA and protein factors) is responsible for the cardioprotective and regenerative effects of MSC-Exos.

**Table 1 biomolecules-10-00707-t001:** List of components of the mesenchymal stem cell-derived exosomes (MSC-Exos) molecular cargo selected for their known potential to regulate the angiogenesis process.

MSC-Exo Molecular Cargo Component	Function	Reference
Avian reticuloendothelial virus oncogene homolog A	RELA, along with p50, is a constituent of the NF-κB heterodimer that mediates NF-κB gene transactivation activity, which includes numerous angiogenesis-related genes [134].	[119]
Cadherin	Vascular endothelial cadherin modulates angiogenesis and the structural integrity of blood vessels [135].	[119]
Epidermal growth factor receptor	The EGFR signal transduction pathway regulates angiogenesis and is especially pro-angiogenic during tumorigenesis [136].	[119]
Extracellular matrix metalloprotease inducer	EMMPRIN mediates cell migration and angiogenesis upstream of VEGF and MMP-9 [121]. EMMPRIN promotes angiogenesis by directly elevating the expression of VEGF [137].	[128]
Ezrin	Ezrin plays a key role in the actin-based cellular functions required for cell locomotion that are important in angiogenesis [138].	[130]
Fibroblast growth factor	FGF is a potent inducer of angiogenesis via its mitogenic action on vascular and capillary endothelial cells. Specifically, it achieves this by driving VEGF-induced angiogenesis [139,140].	[119]
Galectin-1	Galectin-1 contributes to multiple steps of the angiogenesis pathway; pro-angiogenic signalling via VEGF receptors and H-Ras is augmented by galectin-1 [141].	[130]
Interleukin-8	Chemokine IL-8 exerts potent angiogenic properties on ECs through interaction with the receptors C-X-C chemokine receptor type (CXCR1) and CXCR2 [142].	[129]
Platelet-derived growth factor	PDGF is heavily involved in the angiogenic processes in a vast array of physiological contexts. PDGF interacts with different PDGFRs, which in turn activate multiple pro-angiogenic pathways such as the MAPK and PI3K pathways [143,144].	[129]
Platelet-derived growth factor receptors	PDGFs interact with PDGFRs to activate the pro-angiogenic MAPK and PI3K signalling pathways [144].	[129]
p195	p195 functions to link VEGFR2 to the vascular endothelial cadherin-containing adherens junctions, thereby promoting VEGF-stimulated angiogenesis [145].	[130]
Nuclear factor-κb	NF-κB is a transcription factor highly associated with tumour angiogenesis. It activates numerous pro-angiogenic genes such as VEGF, IL-8, and several MMPs [146,147].	[119]
Transforming growth factor-β	TGF-β induces angiogenesis through its binding to TGF-β receptor complexes present on ECs [148]. The subsequent signalling response is highly context-dependent: it can result in promotion or suppression of endothelial migration, proliferation, permeability, and sprouting [149].	[106,129]
Vascular endothelial growth factor	VEGF is an important key factor involved in maintaining vascular homeostasis and stimulating the angiogenic cascade [150,151].	[128,129]
miR-21	miR-21 activates the PTEN/VEGF signalling pathway after acute MI to exert cardioprotective pro-angiogenic effects [152].	[110,133]
miR-21-5p	miR-21-5p leads to increased expression of the TGF-β signalling pathway, pro-angiogenic VEGF-α, and angiopoietin-1, and ANP and BNP [132].	[132]
miR-23	miR-23 interact with Sprouty2, Sema6A, and Sema6D in ECs to induce sprouting angiogenesis [153].	[133]
miR-23a-3p	Hypoxic tumour exosomal miR-23a directly targets prolyl hydroxylase 1 and 2 (PHD1 and 2) in endothelial cells, promoting tumour angiogenesis [154].	[133]
miR-1246	Colon tumour exosome miR-1246 has been found to promote angiogenesis via Smad 1/5/8 signalling in ECs [155].	[133]

**Table 2 biomolecules-10-00707-t002:** List of components of the mesenchymal stem cell-derived exosomes (MSC-Exos) molecular cargo selected for their known potential to regulate the apoptosis process.

MSC-Exo Molecular Cargo Component	Function	Reference
miR-19a	miR-19a downregulates PTEN and BIM expression resulting in AKT and ERK signalling pathways activation while inhibiting JNK/caspase-3 activation by targeting the transcription factor SOX-6 [179].	[179]
miR-21	miR-21 is involved in several intracellular signalling pathways and modulates apoptotic proteins in CMCs, such as PDCD4, TLR4, NF-κB, and PTEN/AKT/Bcl-2 [167]. In addition, miR-21 is involved in PTEN/PI3K/AKT pathway modulation [175].	[110,123,133]
miR-22	miR-22 inhibits apoptosis by targeting Mecp2 [173].	[173]
miR-24	miR-24 represses BIM translation to suppress apoptosis [185].	[110,176]
miR-125b-5p	miR-125b-5p protects ECs from apoptosis and necrosis under oxidative stress via interaction with SMAD4 [186].	[131]
miR-144	miR-144 counteracts apoptosis in hypoxic CMCs by interacting with the PTEN/PI3K/AKT pathway [187,188]. Conversely, miR-144 is also known for suppressing proliferation and promoting apoptosis in tumours.	[166]
miR-221	miR-221 inhibits PUMA, a pro-apoptotic member of the Bcl-2 protein family [181]	[181]
miR-451	miR-451 modulates the TLR4/NF-κB pathway, resulting in a significant apoptosis reduction [189].	[179]
miR-486-5p	miR-486-5p represses the PTEN pathway while activating the PI3K/AKT pathway in CMCs to prevent apoptosis [177].	[167]
lncRNA-NEAT1	lncRNA-NEAT1 inhibits miR-142-3p, which is known to induce apoptosis and cardiac dysfunction [184]. Additionally, activation of the lncRNA-NEAT1/miR-142 axis enhances FOXO1 activity in CMCs, resulting in apoptosis gene expression modulation.	[184]

**Table 3 biomolecules-10-00707-t003:** List of components of the mesenchymal stem cell-derived exosomes (MSC-Exos) molecular cargo selected for their known potential to regulate the immune response.

MSC-Exo Molecular Cargo Component	Function	Reference
Galectin-1	Galectin-1 functions as a homeostatic agent by modulating innate and adaptive immune responses [233]. Galectin-1 inhibits cell growth, induces cell cycle arrest, and promotes apoptosis of activated immune cells [234,235,236].	[130,214]
Programmed death-ligand 1	PD-L1 is a crucial part of the programmed death-1 (PD-1)/PD-L1 pathway, which regulates T cell responses and its effects on immunological tolerance and immune-mediated tissue damage [237].	[214]
Transforming growth factor-β	TGF-β is a potent cytokine having effects on many different cells in the immune system (including T cells and dendritic cells) and exerting both pro- and anti-inflammatory effects depending on the context in which it is acting [238].	[106,129,214]
miR-21	miR-21 acts as a negative regulator of T cell activation by targeting guanine nucleotide-binding protein G subunit alpha (GNAQ), pleckstrin homology domain-containing family A member 1 (PLEKHA1), and CXCR4 [239]. Mature miR-21 regulates the anti-inflammatory responses and polarises macrophages to the M2 phenotype [240].	[110,123,133,223]
miR-22	miR-22 ameliorates fibrosis and improves cardiac function post-MI [241].	[173]
miR-24	miR-24 limits aortic vascular inflammation through interaction with CHI3L-1, which itself is a regulator of inflammation and tissue remodelling [216].	[110,176,216]
miR-29	miR-29 reduces fibrosis via repression of several collagen genes [215].	[110]
miR-34a-5p	miR-34a-5p is a central regulator of NF-κB in T cells [234,242] and differentiation towards M2 macrophage polarisation [243].	[223]
miR-133	miR-133 ameliorates fibrosis and improves cardiac function post-MI [244].	[232]
miR-146a-5p	miR-146a can contribute towards M1 to M2 polarisation by downregulating M1-marker genes [245].	[223]
miR-181a	miR-181a inhibits the inflammatory response through interaction with the c-Fos protein, a key immunoactivator that contributes to dendritic cell-related immune functions [217].	[217]
miR-182	miR-182 interacts with the TLR4/NF-κB/PI3K/AKT pathway, regulate regulator of macrophage polarisation [218].	[218]

**Table 4 biomolecules-10-00707-t004:** Summary of the clinical trials involving MSC-extracellular vesicles (EVs).

Disease	Study Type	Phase	Trial ID	Reference
Bronchopulmonary Dysplasia	Interventional	Phase I	NCT03857841	[221,248]
Dystrophic Epidermolysis Bullosa	Interventional	Phase I/IIA	NCT04173650	[249]
Acute Ischemic Stroke	Interventional	Phase II	NCT03384433	[250,251,252]
Dry Eye	Interventional	Phase II	NCT04213248	[253,254,255,256]
Macular Holes	Interventional	Phase I	NCT03437759	[257,258]
Pancreatic Adenocarcinoma	Interventional	Phase I	NCT03608631	[259,260,261,262]
Diabetes Mellitus Type 1	Interventional	Phase III	NCT02138331	[263,264]

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
