# Peer review of "Novel Applications of Mesenchymal Stem Cell-Derived Exosomes for Myocardial Infarction Therapeutics"

_biomolecules, 2020, doi:10.3390/biom10050707_

Round 1
Reviewer 1 Report
In this paper, the authors review the potential applications of mesenchymal stem cells derived exosomes for myocardial infarction therapeutics. The manuscript is well written, informative and the information is presented in a logical manner. However, there are several issues that need to be addressed:
- Page 2, line 58 “MSC-Exorived exosomes” means MSC-derived exosomes?
- Page 2, line 64 “EVs are prokaryote and higher eukaryote cell-secreted nanosized vesicles (30 nm to several micrometers in diameter)” – the term “nanosized” is incorrect in this context, ass it refers to vesicles smaller than 1 micrometer.
- Page 2 line 76 The authors state that “the vast range of cell-type-specific surface proteins represent an additional layer of complexity to exosome classification.” I believe they refer to EVs, not just exosomes, since these small vesicles have their origin in the intracellular compartment, and they do not result through the budding of the plasmalemma.
- Page 3, lines 106-109, there are many abbreviations which need to be addressed.
- Page 5, line 205, EC abbreviation needs to be explained.
- Pages 6, 10 and 13, Table 1, 2 and 3 are not mentioned in the manuscript text.
- Page 15, Figure 2 is not mentioned in the manuscript text.
- Section 4, Clinical trials involving stem-cell derived exosomes. Although very interesting and informative, it does not have any relation with the topic of the review. The authors mention that the variability in cell culture methods for exosomes production, exosomes isolation, purification and dose for administration are the reason why there are not yet clinical trials for the use of exosomes in myocardial infarction. Therefore, in order to use the information given by the clinical trials using exosomes in other conditions, I suggest stating the aforementioned details (cell type, isolation method etc.) for each trial and then discuss the similarities and differences and how could this information could apply to a clinical trial on heart diseases.
Reviewer 2 Report
The review by Tan et al deals selectively with exosomes derived from MSCs as vehicles for treating cardiovascular disease in particular MI and heart failure. While there have been several recent reviews on the biology and clinical potential of exosomes for heart disease treatment, including by the communicating authors on ischemic heart disease and fibrosis, and there is some overlap, by focusing on MSCs and highlighting the relationships between the exosomal cargo and specific signaling pathways, the present review covers new ground and from a different angle, and is therefore of value. I have the following comments:
General:
From a content perspective and because this review is specifically targeted at MSC exosomes for the heart, the end section of the article that deals with non-cardiac clinical trials should perhaps be shortened into a brief paragraph overview of the most relevant ongoing and approved trials of exosome therapy, in each case identifying the exosome source and purification procedures - principally to make the point that such treatments appear to be safe and are showing some promise in these non-cardiac interventions. It is indeed a tantalizing question why there have been heart failure trials of MSCs and CSCs but not exosomes, given that the latter should be safer and potentially off-the-shelf. Rather or in addition I recommend that the authors have a final paragraph that compares the potential and relative promise of exosomes from different sources as ideal candidate(s) for a first in human clinical trial for human MI and/or heart failure – from safety, source and therapeutic perspectives. For example there is some indication from other studies that adipose MSCs may be the preferable source of MSC-Ex, and additional work including that reviewed by the corresponding authors that cardiac progenitor exosomes and/or those from cardiac derived iPSC may be preferable. This may help clarify this rather murky issue. All of the trials cited by the authors in T4 are using MSCs – is this purely for convenience or because of the inherent and unique immunomodulation properties of MSCs and their Exo cargoes?
There is a tendency for the authors to discuss cardiac regeneration in a general sense (e.g. L. 536-7; L. 540; L. 545; L. 560) – perhaps they should acknowledge/ clarify up front that this relates primarily and maybe exclusively to the vasculature because there is only very slim evidence that cardiac myocytes in aged human or even aged animals hearts retain much/any regenerative potential of physiological significance.
The article needs extensive editing for typos, grammar, and smoother flow. In many cases sentences are clumsy and there is overuse of link words such as “additionally”, “in addition”, “furthermore” that distract from the flow. Please check for appropriate use of “which” vs. “that”.
Specific minor points:
Introduction – should be “necrosis and apoptosis”
L 69 – “pathological and physiological”
L 132-45 – can the authors comment on the use of asymmetric flow field flow fractionation as a preferred purification method
L 141-53 – are all of these markers for MSCs?
L 202 - network organization have been observed in in vitro and in vivo -- ?
L 216 – please comment on how EV activate PKA
L. 230 – ischemic MSCs attempt to -- this suggest a conscious property of the cells – please rephrase
Table 1: PDGF interacts with different PDGFRs, which (that?) in turn stimulate activate multiple -- which is it ?
L. 273-276 – awkward, rephrase – most readers will know what I/R is.
L. 312 – perhaps replace “producing” with “conferring”
L. 322 – is “artificially” the best word for this?
L. 324 – GATA 4 regulates which mRNAs?
Tables 2 and 3 have multiple duplications, please correct. Also I don’t think it unnecessary to re-identify the cargo component (miR/protein) in the middle columns of the Tables.
Suggest cut Table 4 or replace with a Table comparing pros and cons of different exosome sources and optimizations.
Reviewer 3 Report
The review article describes the application of MSC-derived exosomes in the therapy of myocardial infarction. The topic is interesting and the manuscript is well-written.
My comments:
- Some sentences should be backed up by references. In details, sentences in lines 37-39; 39-41; 128; 223-231; 277-281; 458; 539-542.
- Tables 1, 2, and 3 should be referred to in the main text of the relative paragraph.
- In line 58: Should be MSC-derived?
- In section 4: The authors should indicate the keywords used to find the clinical trials on the website.
- The authors describe cell therapies attempted to treat MI, they should detail this section a little more to highlight the limitations that arose from such studies.
- In section 2.1 the authors should include more references and consider adding a table that summarizes the main methods for exosome/EVs isolation and separation,highlighting their advantages and disadvantages.
- Figure 3 is not referred to in the text and should be enlarged. The insets are not clearly visible. This figure should be accompanied by a more descriptive caption.
